# Functionally diverse human T cells recognize non-microbial antigens presented by MR1

Marco Lepore[1], Artem Kalinichenko[1], Salvatore Calogero[1], Pavanish Kumar[2], Bhairav Paleja[2], Mathias Schmaler[1], Vipin Narang[2], Francesca Zolezzi[2], Michael Poidinger[2,3], Lucia Mori[1,2], Gennaro De Libero[1,2]*

[1]Department of Biomedicine, University Hospital and University of Basel, Basel, Switzerland; [2]Singapore Immunology Network, A*STAR, Singapore, Singapore; [3]Singapore Institute for Clinical Sciences, A*STAR, Singapore, Singapore

**Abstract** MHC class I-related molecule MR1 presents riboflavin- and folate-related metabolites to mucosal-associated invariant T cells, but it is unknown whether MR1 can present alternative antigens to other T cell lineages. In healthy individuals we identified MR1-restricted T cells (named MR1T cells) displaying diverse TCRs and reacting to MR1-expressing cells in the absence of microbial ligands. Analysis of MR1T cell clones revealed specificity for distinct cell-derived antigens and alternative transcriptional strategies for metabolic programming, cell cycle control and functional polarization following antigen stimulation. Phenotypic and functional characterization of MR1T cell clones showed multiple chemokine receptor expression profiles and secretion of diverse effector molecules, suggesting functional heterogeneity. Accordingly, MR1T cells exhibited distinct T helper-like capacities upon MR1-dependent recognition of target cells expressing physiological levels of surface MR1. These data extend the role of MR1 beyond microbial antigen presentation and indicate MR1T cells are a normal part of the human T cell repertoire.

*For correspondence: gennaro.delibero@unibas.ch

## Introduction

T lymphocytes can detect a diverse range of microbial and self-antigens, including lipids presented by non-polymorphic CD1 molecules (*Zajonc and Kronenberg, 2007*), and phosphorylated isoprenoids bound to Butyrophilin 3A1 (*Sandstrom et al., 2014*; *Vavassori et al., 2013*), which stimulate TCR Vγ9-Vδ2 cells, the major population of TCR γδ cells in human blood. The heterogeneous phenotypic and functional properties of these T cells support specialized roles in host protection against infections, cancers, and autoimmunity (*Cohen et al., 2009*; *Mori et al., 2016*; *Tyler et al., 2015*). The repertoire of T cells specific for non-peptide antigens was expanded to include mucosal associated invariant T (MAIT) cells, which are restricted to MHC I-related protein MR1 and were originally described within mouse gut (*Treiner et al., 2003*). Elegant functional studies showed that MAIT cells react to different bacteria by recognizing non-proteinaceous antigens (*Gold et al., 2010*; *Le Bourhis et al., 2010*). Finally, the stimulatory antigens were identified as small riboflavin precursors produced by a wide range of yeasts and bacteria (*Corbett et al., 2014*; *Kjer-Nielsen et al., 2012*; *Soudais et al., 2015*). MAIT cells display one of three alternative semi-invariant TCRs in which the TRAV1-2 gene is rearranged with genes encoding TRAJ33, J12 or J20 (*Gold et al., 2014*; *Lepore et al., 2014*; *Reantragoon et al., 2012*), and is combined with an oligoclonal TCR-$\beta$ repertoire (*Lepore et al., 2014*). Cells expressing these evolutionary-conserved TCRs are frequent in human blood, kidney and intestine, and comprise a major fraction of T cells resident in the liver (*Dusseaux et al., 2011*; *Lepore et al., 2014*; *Tang et al., 2013*). Following activation, MAIT cells

**eLife digest** White blood cells called T cells recognize germs and infected cells, and get rid of other cells in the body that look different to healthy cells – for example, tumor cells. These activities all depend on a molecule called the T cell receptor (or TCR for short), which is found on the surface of the T cells. Each TCR interacts with a specific complex on the surface of the target cell. One of the molecules recognized by the TCR is known as MHC class I-related (shortened to MR1). This molecule attracts TCRs to infected cells, but it was not know if the MR1 molecule could attract TCRs to cancer cells too.

Lepore et al. now show that there are indeed T cells in humans that recognize cancer cells through interaction with the MR1 molecules produced by the cancer cells. This new group of T cells has been named MR1T, and the cells can be easily detected in the blood of healthy individuals. The cells can be classified as a new cell population based on their capacity to recognize MR1 and how they react with different types of cancer cells.

Importantly, the MR1 that attracts these TCRs is the same in all people, and so the same TCR may recognize MR1-expressing cancer cells from different patients. The next challenge is to identify MR1T cells that recognize and kill cancer cells from different tissues. These studies will hopefully pave the way for new and broader strategies to combat cancer.

release an array of pro-inflammatory and immunomodulatory cytokines, and can mediate direct killing of microbe-infected cells (*Dusseaux et al., 2011*; *Kurioka et al., 2015*; *Le Bourhis et al., 2013*; *Lepore et al., 2014*; *Tang et al., 2013*). MAIT cells react to a broad range of microbes, but it remains unknown whether the role of MR1 extends beyond presentation of microbial metabolites to MAIT cells.

MR1 is a non-polymorphic MHC I-like protein that is expressed at low to undetectable levels on the surface of many cell types (*Huang et al., 2008*; *Miley et al., 2003*). Following bacterial infection, antigen-presenting cells (APC) up-regulate surface expression of MR1 due to protein stabilization upon antigen binding (*Huang et al., 2008*; *McWilliam et al., 2016*; *Miley et al., 2003*). MR1 is highly conserved across multiple species, with human and mouse MR1 sharing >90% sequence homology at the protein level (*Riegert et al., 1998*). To date, the MR1-bound ligands reported to simulate MAIT cells include three ribityl lumazines (RLs), microbial pterin-like compounds displaying a ribitol at carbon 8 (*Kjer-Nielsen et al., 2012*), and two unstable adducts formed by non-enzymatic reaction of microbial 5-ribityl amino uracil (5-RAU) with either host-derived or bacterial carbonyls such as glyoxal or methyl-glyoxal (*Corbett et al., 2014*). RLs bind to MR1 via hydrogen bonds and hydrophobic interactions with amino acids in the antigen binding cleft (*Kjer-Nielsen et al., 2012*; *Patel et al., 2013*), whereas binding of unstable 5-RAU-derivatives requires covalent trapping within the MR1 pocket via formation of a Schiff base with Lysine 43 (*Corbett et al., 2014*; *Patel et al., 2013*). While three additional molecules have also been demonstrated to bind MR1 via the same mechanism, including 6-formyl-pterin (6-FP), acetyl-6-formyl-pterin (Ac-6-FP), and acetylamino-4-hydoxy-6-formylpteridine dimethyl acetal (*Kjer-Nielsen et al., 2012*; *Patel et al., 2013*; *Soudais et al., 2015*), these compounds lack the ribityl moiety necessary for MAIT cell activation and instead inhibit stimulation by microbial ligands (*Corbett et al., 2014*; *Kjer-Nielsen et al., 2012*; *Soudais et al., 2015*). A recent study reported that 6-FP and Ac-6-FP can also stimulate a second population of 'atypical' MR1-restricted T cells of unknown function in the blood of healthy human donors (*Gherardin et al., 2016*), but this population, did not express the canonical TRAV1-2 gene which defines MAIT cells. Together, these findings indicate a degree of MR1 plasticity in accommodating diverse chemical structures and suggest that other MR1-restricted T cell populations may exist in vivo.

Here we report a novel population of MR1-restricted T cells (hereafter termed MR1T cells) that is readily detectable in blood from healthy individuals. MR1T cells express diverse TCRα and $\beta$ genes and were not able to recognize previously identified microbial or folate-derived ligands of MR1. Instead, MR1T cells recognized self-antigens presented by MR1 and expressed a broad selection of cytokines and chemokine receptors. Functionally, MR1T cells were capable of inducing dendritic cell

(DC) maturation and promoted innate defense in intestinal epithelial cells. These findings demonstrate that MR1 can present non-microbial antigens to a novel population of functionally diverse human T cells with potentially wide-ranging roles in human immunity.

## Results

### MR1-reactive T cells are readily detectable in healthy individuals

During our previous study on the repertoire of human MAIT cells (*Lepore et al., 2014*), we detected an atypical MR1-restricted T cell clone that did not react to microbial ligands. This T cell clone (DGB129) recognized cell lines constitutively displaying surface MR1 (CCRF-SB lymphoblastic leukemia cells, or THP-1 monocytic leukemia cells; *Figure 1A*) or A375 melanoma cells (A375-MR1) transfected with an MR1-$\beta$2m fusion gene construct (*Lepore et al., 2014*) (*Figure 1A*) in the absence of any exogenously added antigens (*Figure 1B*). Sterile recognition of MR1$^+$ target cells was fully inhibited by blocking with anti-MR1 monoclonal antibodies (mAbs) (*Figure 1B*), and thus resembled the MAIT cell response to *E. coli*-derived antigens assessed in parallel (*Figure 1C*). Importantly, DGB129 T cells also failed to recognize the synthetic MAIT cell agonist 6,7-dimethyl-8-D-ribityllumazine (RL-6,7-diMe; *Figure 1D*) (*Kjer-Nielsen et al., 2012*), differently from a control MAIT cell clone, which instead was stimulated in MR1-dependent manner by this compound (*Figure 1E*). Unlike the canonical semi-invariant TCR typical of MAIT cells, DGB129 cells displayed a TCR$\alpha\beta$ heterodimer comprising TRAV29/DV5 rearranged to TRAJ23 and TRBV12-4 rearranged to TRBJ1-1. Expression of these TCR$\alpha$ and $\beta$ genes in a TCR-deficient cell line (SKW-3 T lymphoblastic leukemia) conferred MR1 reactivity in the absence of exogenous antigens comparable to that displayed by DGB129 cells (*Figure 1F*), while in control experiments, transduction of TCR$\alpha$ and $\beta$ genes of a representative

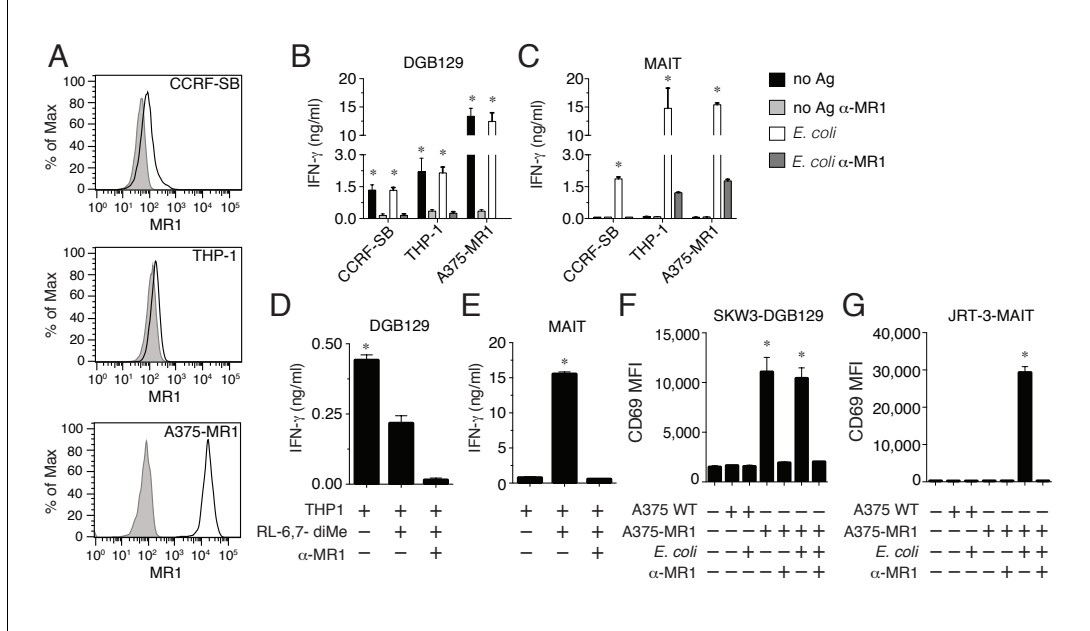

**Figure 1.** Recognition of non-microbial antigens by MR1-restricted T cells. (A) Surface expression of MR1 by CCRFSB, THP-1 and A375-MR1 cells. Grey histograms indicate staining with isotype-matched control mAbs. Stimulation of (B) T cell clone DGB129 or (C) MAIT cell clone SMC3 by the three cell lines in A in the absence (no Ag) or presence of *E. coli* lysate (*E. coli*) and/or anti-MR1 blocking mAbs ($\alpha$-MR1). Columns indicate IFN-$\gamma$ release (mean + SD). Stimulation of (D) DGB129 MR1T or (E) SMC3 MAIT cells by THP-1 cells, constitutively expressing surface MR1, loaded with synthetic 6,7-dimethyl-8-D-ribityllumazine (RL-6,7-diMe) with or without anti-MR1 mAbs. Columns indicate mean IFN-$\gamma$ release + SD. Stimulation of (F) SKW-3 cells expressing the DGB129 TCR (SKW3-DGB129) or (G) J.RT3-T3.5 cells expressing the MAIT MRC25 clone TCR (J.RT3-MAIT) with A375 cells that expressed (A375-MR1) or lacked (A375-WT) MR1, with or without *E. coli* lysate and/or anti-MR1 mAbs. CD69 median fluorescence intensity (MFI) ± SD of duplicate cultures of transduced T cells are shown. The CD69 MFI of transduced T cells cultured in the absence of APCs is also shown. Data are representative of four (A, B and C), two (D and E), and three (F and G) independent experiments. *p<0.05 (Unpaired Student's t-test).

MAIT cell clone conferred the ability to recognize the same target cells in MR1-dependent manner only in the presence of *E. coli* antigens (*Figure 1G*). Collectively, these data highlight a critical role of the TCR in mediating DGB129 cell recognition of MR1-expressing APCs and suggest that MR1 can present non-microbial antigens to T cells other than MAIT cells.

To investigate the presence of these unpredicted MR1-restricted T cells in different individuals, we performed combined in vitro stimulation and single cell cloning experiments using total blood T cells. Purified T cells from two healthy donors were labeled with the proliferation marker CellTrace violet (CTV) and stimulated with irradiated A375-MR1 cells in the absence of exogenous antigens. The choice of A375-MR1 cells relied on their potent capacity of inducing sterile DGB129 T cell activation (*Figure 1B*) and their high expression of surface MR1 molecules (*Figure 1A*), which we reasoned could maximize the chance of stimulating and thus expanding DGB129-like MR1-restricted T cells present in the blood. Amongst the T cells of both donors, we observed a significant fraction of proliferating cells that expressed high levels of the activation marker CD137 following re-challenge with A375-MR1 cells. These activated T cells were sorted and cloned by limiting dilution (*Figure 2A*). Individual T cell clones were then interrogated for their capacity to recognize A375-MR1 and A375 cells lacking MR1 (A375-WT). In both donors we found that a major fraction of T cell clones (126/195 and 37/57, respectively) displayed specific recognition of A375-MR1 cells (*Figure 2B,D*), which was potently inhibited by anti-MR1 blocking mAbs (*Figure 2C,E*). Staining with TCR V$\beta$-specific mAbs of 12 MR1-reactive T cell clones revealed that they expressed seven different TRBV chains (TRBV4-3, 6-5/6-6/6-9, 9, 18, 25–1, 28, 29–1) with some of the clones sharing the same TRBV gene. Furthermore, none expressed the TRAV1-2 chain, canonical for MAIT cells. These data suggested that MR1-restricted T cells other than MAIT cells exist in the blood of healthy donors, can express diverse TCRs and are able of clonal expansion following in vitro stimulation in the absence of microbial ligands.

Lack of specific markers did not allow univocal identification of these novel T cells ex vivo by standard flow cytometry. Therefore their frequency was estimated by combining flow cytometry analysis after very short-time in vitro stimulation and single T cell cloning experiments. Purified blood T cells from five healthy donors were co-cultured overnight with A375 cells either lacking or over-expressing MR1 and analyzed for the expression of the activation markers CD69 and CD137 (*Figure 3A*). In all of the five donors screened, the percentage of CD69$^{high}$CD137$^+$ T cells detected was consistently higher after stimulation with A375-MR1 cells (range 0.034–0.072% of T cells) than after co-culture with A375-WT cells (range 0.015–0.032%) (*Figure 3A,B*). As the two types of APCs differed for MR1 expression, we assumed that MR1-reactive T cells might account for the increased numbers of activated T cells after stimulation with MR1-positive APCs. Using this approach, we estimated that the circulating T cell pool of the analyzed individuals contained A375-MR1-reactive T cells at frequency ranging between 1:2500 (0.072–0.032 = 0.04%) and 1:5000 (0.034–0.015 = 0.019%). This estimated frequency, although representing an approximation, is within the range of the frequency of peptide-specific CD4$^+$ T cells after antigen exposure (*Lucas et al., 2004*; *Su et al., 2013*). We also performed parallel experiments in which overnight-activated CD69$^{high}$CD137$^+$ T cells (0.065%) were sorted from one of these donors (Donor C, *Figure 3A*, right panel) and were cloned. Indeed, 31 out of 96 screened T cell clones (32%) displayed specific reactivity to A375-MR1 cells (*Figure 3C*), which was inhibited by anti-MR1 mAbs (*Figure 3D*). Accordingly, we calculated that the frequency of A375-MR1-responsive T cells among blood T cells of this donor was 1:5000 (0.065 $\times$ 0.32 = 0.02%), a value consistent with the previously estimated range. Detailed analysis of representative T cell clones derived from three donors confirmed that they displayed diverse TCR$\alpha$ and $\beta$ chains and indicated differential expression of CD4, CD8 and CD161 (*Table 1*).

Collectively, these findings suggested that identified MR1-responsive T cells are a novel yet readily detectable polyclonal population of lymphocytes in the blood of healthy human individuals (hereafter termed MR1T cells).

## MR1T cells recognize antigens bound to MR1

We next studied the basis of MR1T cell reactivity. Firstly, we tested whether the MR1T cell clones could recognize microbial antigens, in analogy to MAIT cells. While a control MAIT cell clone reacted to A375-MR1 cells only in the presence of *E. coli* lysate, activation of different MR1T cell clones was not enhanced by the *E. coli* lysate (*Figure 4A*). Consistent with these data, MR1-negative A375-WT cells failed to stimulate either type of T cell, irrespective of whether *E. coli* lysate was added

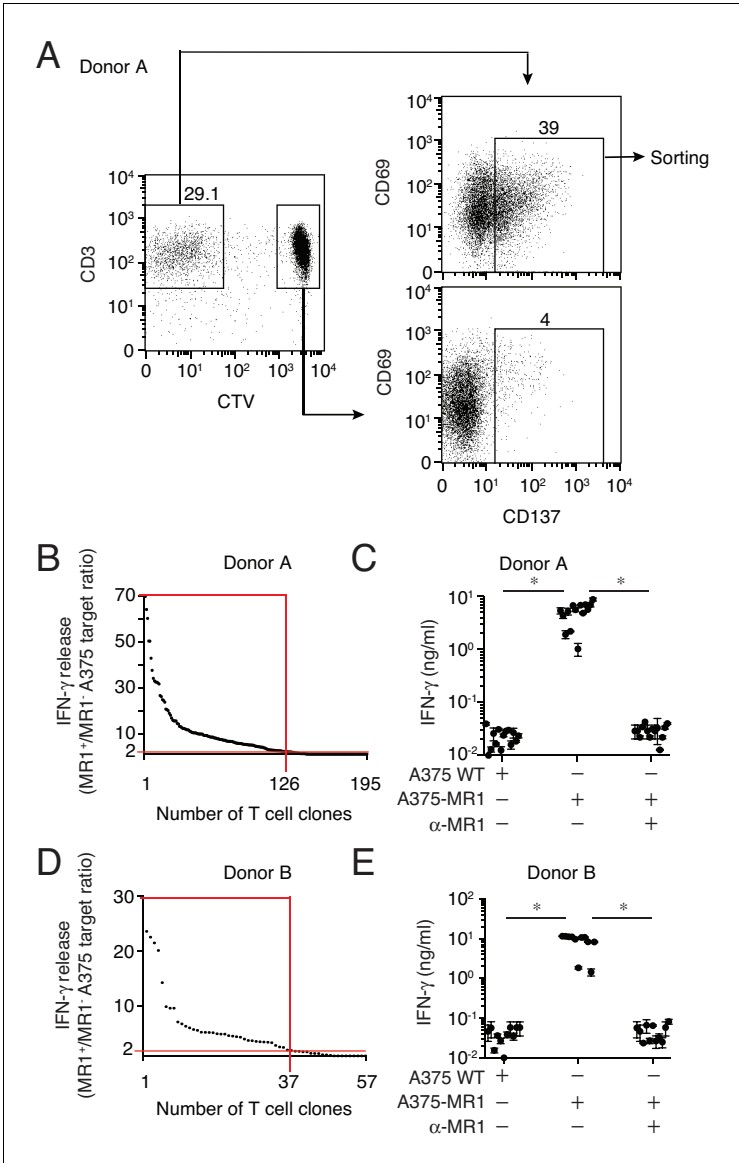

**Figure 2.** Isolation of non-MAIT MR1-restricted T cell clones after stimulation with A375-MR1 cells in the absence of microbial antigens. (**A**) FACS analysis of purified T cells previously expanded with irradiated A375-MR1 cells following overnight co-culture with A375-MR1 cells. Left dot plot shows CD3 and CellTrace violet (CTV) staining in live cells. Upper right and bottom right dot plots show CD69 and CD137 expression on CD3+CTV− and CD3+CTV+ gated cells, respectively. Arrows indicate gating hierarchy. Numbers indicate the percentages of cells within the gates. Cells from Donor A are illustrated as a representative donor. (**B, D**) Cumulative results of T cell clones screening from Donors A and B. T cell clones were generated from CD3+CTV−CD137high sorted T cells as depicted in A. Graphs show the individual clones (x axis) and their IFN-γ release (y axis), expressed as ratio between the amount of cytokine secreted in response to A375-MR1 cells *vs.* A375 WT cells. Each dot represents a single T cell clone, tested at the same time in the indicated experimental conditions. The horizontal red line marks the arbitrary IFN-γ ratio cut-off of two, above which MR1-dependent T cell clone reactivity was set. The intercept of the vertical red line indicates the number of MR1-restricted T cell clones in each donor. Red boxes highlight T cell clones whose reactivity was MR1-dependent. Results are representative of two independent experiments. (**C, E**) IFN-γ release by 14 representative clones from Donor A and 11 clones from Donor B after stimulation with A375 WT, A375-MR1 and A375-MR1 in the presence of blocking anti-MR1 mAbs (α-MR1). Dots represent the IFN-γ release (mean ± SD of duplicate cultures) by each clone. Results are representative of three independent experiments. *p<0.05 (Unpaired Student's t-test).

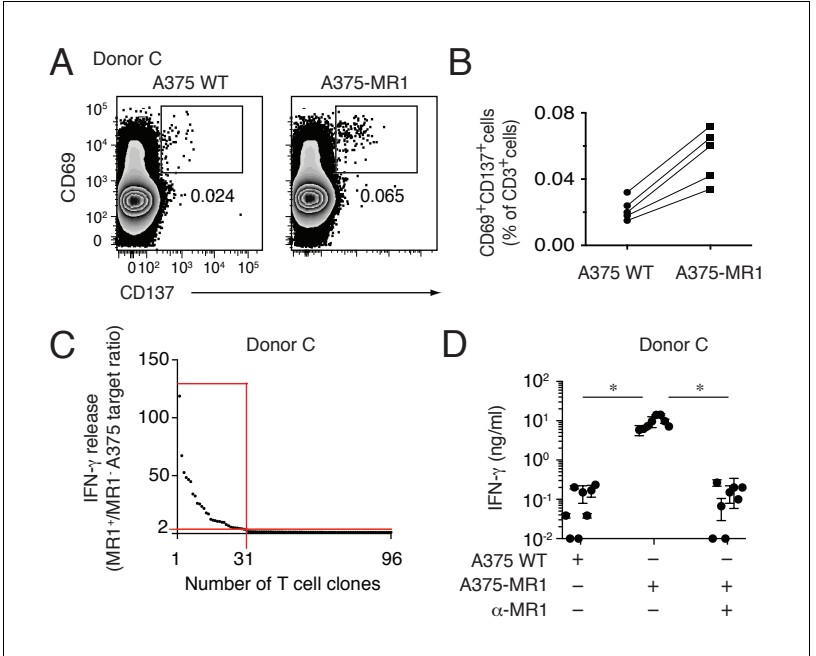

**Figure 3.** Non-MAIT MR1-restricted T cells are readily detectable in the blood of healthy individuals. (**A**) Flow cytometry analysis of purified T cells from a representative donor (Donor C) after overnight co-culture with A375 WT or A375-MR1 cells. Dot plots show CD69 and CD137 expression on live CD3$^+$ cells. Numbers indicate the percentage of cells in the gates. (**B**) Frequency of CD69$^+$CD137$^+$ T cells from five different donors after overnight co-culture with A375 WT or A375-MR1 cells. (**C**) Cumulative results of T cell clone stimulation assays from Donor C. T cell clones were generated from CD3$^+$CD69$^+$CD137$^+$ sorted T cells as depicted in A, right dot plot. The graph shows the number of tested clones (x axis) and IFN-γ release (y axis) expressed as ratio between the amount of cytokine secreted in response to A375-MR1 cells *vs.* A375-WT cells. Each dot represents a single T cell clone, tested at the same time in the indicated experimental conditions. The horizontal red line marks the arbitrary IFN-γ ratio threshold of two above which MR1-dependent T cell clone reactivity was set. The intercept of the vertical red line indicates the number of MR1-restricted T cell clones. Red box highlights T cell clones whose reactivity was MR1-dependent. Results are representative of two independent experiments. (**D**) Recognition of A375-MR1 but not A375 WT cells in the absence of exogenous antigens by eight representative MR1-restricted T cell clones from Donor C. Inhibition of T cell clone reactivity to A375-MR1 cells by blocking anti-MR1 mAbs (α-MR1). Dots represent the IFN-γ release (mean ± SD of duplicate cultures) by each clone tested in the three experimental conditions. Results are representative of three independent experiments. *p<0.05 (Unpaired Student's t-test).

(*Figure 4A*) and importantly, anti-MR1 mAbs efficiently blocked both MR1T and MAIT cell responses

**Table 1.** Phenotype and TCR gene usage of selected MR1-reactive T cell clones.

| Clone | CD4 | CD8α | TCRα | TCRβ | CD161 |
|---|---|---|---|---|---|
| DGB129 | − | + | TRAV29 | TRBV12-4 | − |
| DGB70 | − | − | TRAV5 | TRBV28 | − |
| DGA28 | − | + | TRAV25 | TRBV29-1 | + |
| DGA4 | − | − | TRAV1-2 | ND | + |
| JMA | − | + | TRAV27 | TRBV25-1 | − |
| TC5A87 | − | + | TRAV13-1 | TRBV25-1 | − |
| CH9A3 | − | + | TRAV24 | TRBV5-5 | − |

ND, not determined.

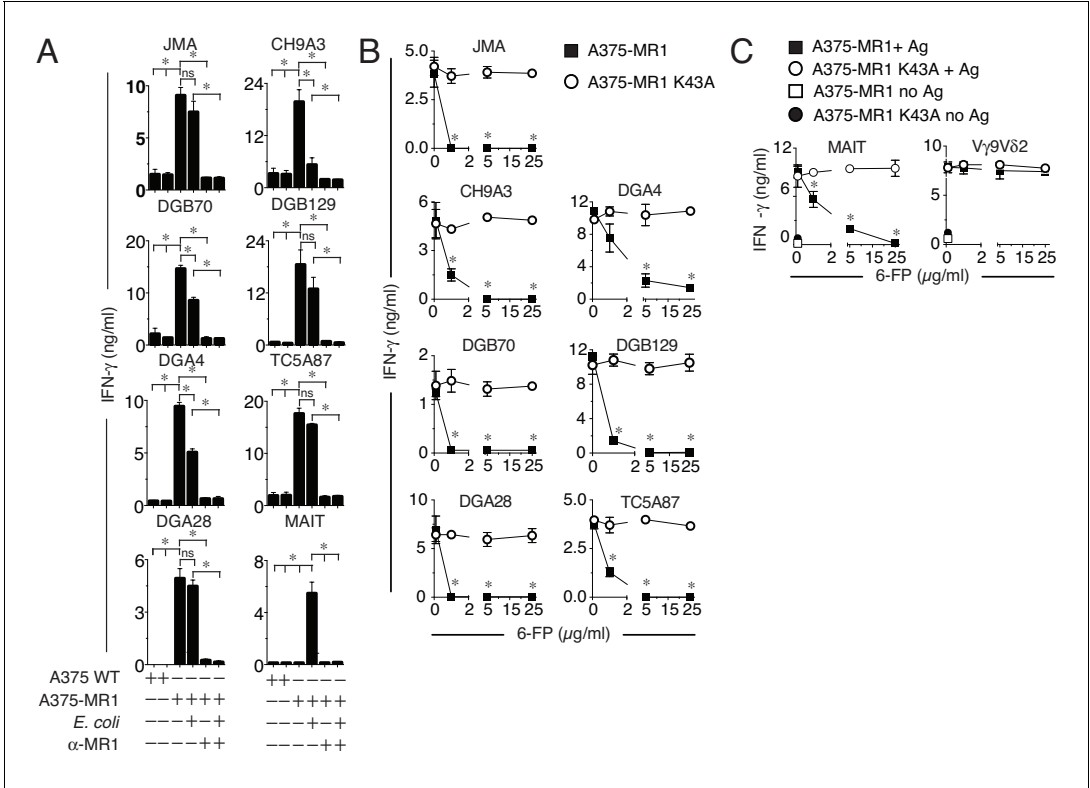

**Figure 4.** MR1T cell clones do not react to microbial ligands or to 6-FP. (**A**) Response of seven MR1T cell clones and one control MAIT cell clone co-cultured with A375 cells expressing (A375-MR1) or not (A375 WT) MR1 in the presence or absence of *E. coli* lysate. Blocking of T cell clone reactivity by anti-MR1 mAbs (α-MR1) is also shown. (**B**) Response of MR1T cell clones to A375 cells expressing either WT MR1 molecules (A375-MR1) or K43A-mutated MR1 molecules (A375-MR1 K43A) in the presence of 6-formyl pterin (6-FP). (**C**) Stimulation of control MAIT cell clone MRC25 or control TCR Vγ9Vδ2 clone G2B9 with A375-MR1 or A375-MR1 K43A cells previously incubated with or without *E. coli* lysate or zoledronate, respectively, either in the absence or presence of 6-FP. Results are expressed as mean ± SD of IFN-γ measured in duplicate cultures. Results are representative of three independent experiments. *p<0.05 (Unpaired Student's t-test).

The following figure supplement is available for figure 4:

**Figure supplement 1.** MR1T cell clones do not recognize Ac-6-FP.

(*Figure 4A*). These findings confirmed that microbial ligands present in *E. coli* and stimulating MAIT cells do not stimulate the tested MR1T cells.

We then tested the response of MR1T cells to the known MR1 ligands 6-FP and Ac-6-FP, which have previously been reported to stimulate a rare subset of TRAV1-2-negative T cells (*Gherardin et al., 2016*) and inhibit MAIT cell activation by microbial antigens (*Corbett et al., 2014*; *Kjer-Nielsen et al., 2012*; *Soudais et al., 2015*). MR1T cell stimulation was impaired in the presence of 6-FP or Ac-6-FP ligands, which also impaired *E. coli* stimulation of control MAIT cells, but did not disrupt TCR γδ cell responses to cognate antigen presented by the same APCs (*Figure 4B,C* and *Figure 4—figure supplement 1A–C*). Notably, 6-FP or Ac-6-FP failed to inhibit the activation of MR1T cells or MAIT cells when the target A375 cells were transduced to express mutant MR1 molecules with defective ligand binding capacity (blockade of Schiff base formation with ligands by mutation of Lysine 43 into Alanine, A375-MR1 K34A; *Figure 4B,C* and *Figure 4—figure supplement 1D, E*). The specific inhibition observed with 6-FP or Ac-6-FP indicated that MR1T cells (i) do not recognize 6-FP and Ac-6-FP; (ii) react to MR1-bound cellular antigens; (iii) are stimulated by ligands that do not require the formation of a Schiff base with MR1.

To gain information on the origin of the recognized antigens we firstly asked whether the stimulatory capacity of target cells was dependent on culture medium constituents, as some MR1 ligands, *e.g.* 6-FP, may derive from folate present in RPMI 1640 medium used for cell culture (*Kjer-*

*Nielsen et al., 2012*). Both THP-1 and A375-MR1 cells were extensively washed and cultivated 4 days in phosphate buffered saline solution (PBS) supplemented exclusively with 5% human serum. Cells were washed daily before being used to stimulate DGB129 MR1T cells and the T cell activation assays were performed in PBS. Both THP-1 and A375-MR1 cells grown in RPMI 1640 or in PBS showed the same stimulatory capacity (*Figure 5A,B*), thus indicating that medium constituents are not responsible for MR1T cell activation. To directly investigate whether the stimulatory antigens were present in target cells, we performed T cell activation assays using as source of antigen two types of lysates. The first lysate was obtained from in vitro cultured THP-1 cells, while the second one was prepared from mouse breast tumors immediately after resection. Two hydrophobic and four hydrophilic fractions were obtained and tested using as APCs THP-1 cells that constitutively express low levels of MR1. The DGB129 clone reacted only to fraction N4, containing highly hydrophilic compounds isolated from both freshly explanted mouse tumor and in vitro cultured THP-1 cells (*Figure 5C,D*). These results ruled out the possibility that stimulatory antigens were derived from RPMI 1640 components and indicated their cellular origin. We also tested the fractions

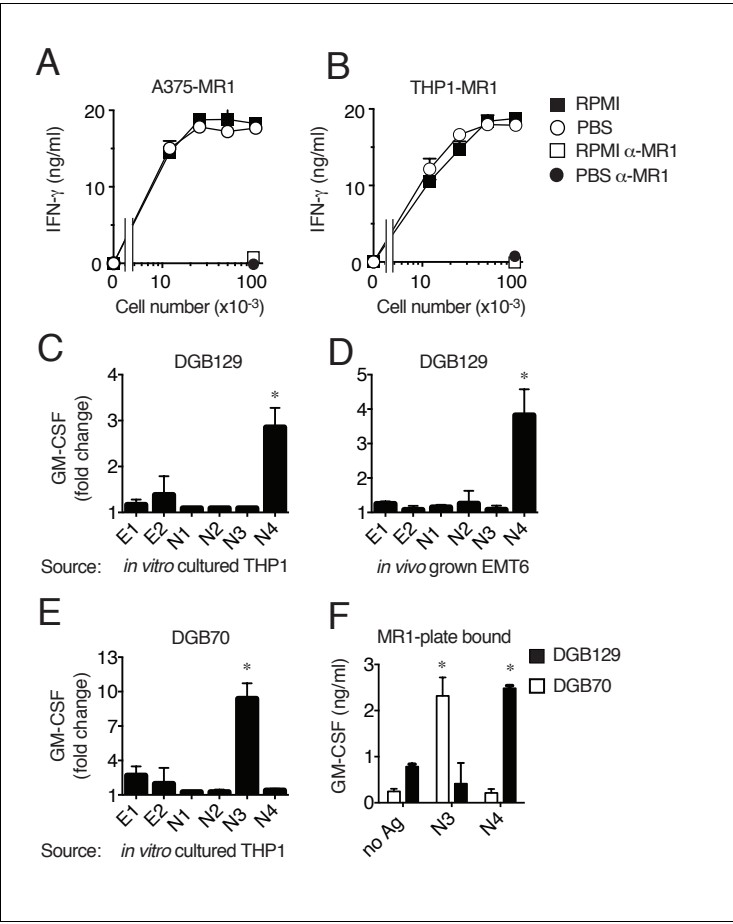

**Figure 5.** MR1T cells recognize diverse antigens not derived from RPMI 1640 medium. Stimulation of the DGB129 MR1T cell clone by MR1-overexpressing (**A**) A375 cells (A375-MR1) and (**B**) THP-1 cells (THP1-MR1) grown for 4 days in RPMI 1640 or in PBS both supplemented with 5% human serum. Inhibition of T cell clone reactivity by anti-MR1 blocking mAbs (α-MR1) is shown. DGB129 cells recognize APCs loaded with fractions isolated from (**C**) THP-1 cell lysate or from (**D**) in vivo grown mouse breast tumor EMT6. Fractions E1 and E2 contain hydrophobic molecules; fractions N1-N4 contain hydrophilic molecules. (**E**) DGB70 MR1T cells react to N3 fraction of THP-1 lysate. (**F**) Stimulation of DGB129 and DGB70 T cells by THP-1-derived fractions N3 and N4 loaded onto plastic-bound recombinant MR1. Shown is T cell release of IFN-γ or GM-CSF mean ± SD of duplicate cultures (representative of three independent experiments). Total cytokine release is shown in panels A, B, F; fold increase over background is shown in panels C, D, E. *p<0.05 (Unpaired Student's *t*-test).

generated from THP-1 lysates with DGB70, another representative MR1T cell clone. DGB70 cells recognized fraction N3 and not N4, (*Figure 5E*), suggesting that at least two distinct compounds differentially stimulated the two MR1T clones. The same fractions were also loaded onto plastic-bound MR1 molecules and showed alternative and specific stimulatory capacity, *i.e.* N3 stimulated only DGB70 cells, while N4 stimulated only DGB129 cells (*Figure 5F*). In the absence of N3 and N4 fractions, the two clones did not react to MR1, further indicating the requirement of specific antigens.

In conclusion, these data indicated that MR1T cells recognize MR1 complexed with ligands not derived from culture medium and present also in tumor cells grown in vivo. At least two diverse molecules were stimulatory, whose nature and structure will be the subject of future studies. As DGB129 and DGB70 clones were isolated from the same donor, these findings also suggested that MR1T cells with different antigen specificities are present in the same individual.

## Differential transcriptomes of antigen-stimulated MR1T cells

We next examined the diversity of MR1T cells by comparing the transcriptional response to antigen stimulation of the two representative MR1T cell clones responding to diverse THP-1 lysate fractions (DGB129 and DGB70). MR1T cells were first rested for three weeks before being stimulated with A375-MR1 cells for 20 hr. The transcriptional profiles of the sorted activated cells (expressing similar levels of both CD25 and CD137 activation markers) were subsequently compared with their unstimulated control counterparts (negative for the two markers) by RNA-sequencing (*Figure 6—figure supplement 1*). Biological replicates were analyzed using stringent cut-off criteria (FDR < 0.05, minimum $\log_2$ fold change >2). The RNA-seq datasets revealed that MR1$^+$ APC stimulated DGB129 cells upregulated 403 genes and downregulated 413 genes, whereas DGB70 cells up-regulated 432 genes and down-regulated 285 genes (*Supplementary file 1*).

We then identified key transcription factors in each clone using global transcription factor gene regulatory network analysis (*Narang et al., 2015*). A subnetwork was created from the identified genes whose expression was modulated upon antigen stimulation, and the key nodes were defined using centrality measures. This approach showed that some master transcription factors were shared between MR1T cell clones, whereas others appeared to be clone-specific (*Figure 6A,B*). Whether assessed for betweenness centrality or tested by the PageRank algorithm, both T cell clone responses were identified as being regulated by genes *TBX21*, *FOXP3*, *FOS*, *RXRA*, *FOSL2*, *IRF4*, *BATF* and *TRIB1*. However, while DGB129 cells were further influenced by expression of *MYC*, *HSP90AB1* and *CREM*, DGB70 cells were instead regulated by *EGR1*, *JUNB*, and *SREBF1* (*Figure 6C*).

## MR1T cells are functionally heterogeneous

We next analyzed the cytokine secretion profile of representative MR1T cell clones upon stimulation by A375-MR1 APCs. All clones tested released IFN-γ (*Figure 7—figure supplement 1A*), probably as a consequence of in vitro expansion (*Becattini et al., 2015*). However, we also observed diverse expression profiles of Th1 (IL-2, TNF-α and TNF-β), Th2 (IL-3, IL-4, IL-5, IL-6, IL-10, IL-13) and Th17 cytokines (IL-17A, G-CSF, GM-CSF), and other soluble factors (MIP-1β, soluble CD40L, PDGF-AA and VEGF; *Figure 7A* and *Figure 7—figure supplement 1B*). The variable combinations and quantities of cytokines expressed by MR1T cells suggested considerable functional plasticity within this population. For example, clone DGA4 secreted large quantities of IL-17A, IL-6, TNF-α and GM-CSF, but failed to secrete the prototypic Th2 cytokines IL-4, IL-5, IL-10 or IL-13, and thus displayed an 'atypical' Th17-like phenotype. In contrast, clone TC5A87 released substantial amounts of VEGF and PGDF-AA, but only little Th1 or Th2 cytokines, and no IL-17A. Notably, four of the seven clones studied (DGB129, CH9A3, DGB70, JMA) displayed a Th2-skewed profile of cytokine release. In the same experiments, both control MAIT cell clones, although derived from two different donors, showed uniform Th1-biased responses following activation by A375-MR1 cells pulsed with *E. coli* lysate. Given that the culture conditions and APCs used in these experiments were identical for all clones, these data indicate that MR1T cells exhibit intrinsic functional heterogeneity.

To further support these findings, we next investigated the expression of three selected chemokine receptors known to be differentially expressed by T cell subsets with distinct functions (*Becattini et al., 2015*) and whose alternative combined expression regulates T cell recirculation and migration to diverse homing sites (*Al-Banna et al., 2014*; *Bromley et al., 2008*; *Thomas et al.,*

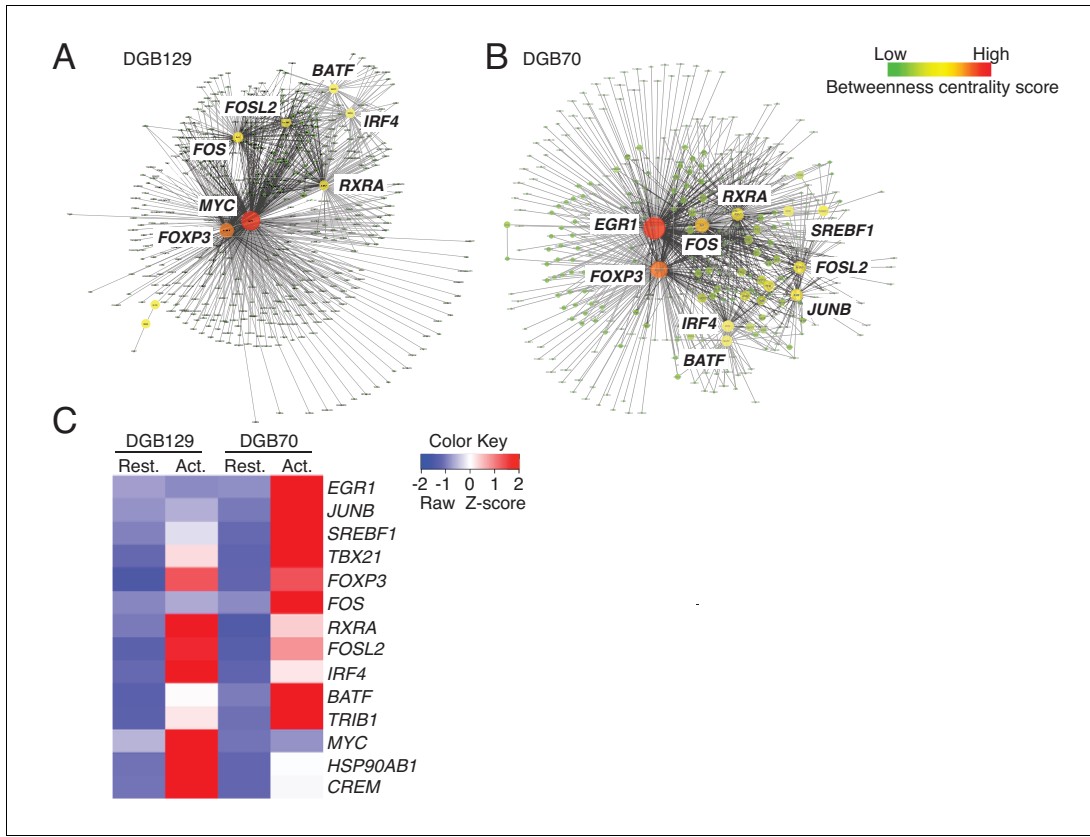

**Figure 6.** MR1T cell clones exhibit divergent transcriptional responses to antigen stimulation. Betweeness centrality analysis illustrating the key transcription factors that were differentially expressed in resting (No Ag) and antigen-activated (Ag Act) MR1T cell clones (**A**) DGB129 and (**B**) DGB70. Color code represents betweeness centrality score. The size of the nodes (or hubs) indicates the relative importance of individual transcription factors within the whole gene network. (**C**) Heat-map comparing key transcription factors that were differentially expressed in resting (Rest.) and antigen-activated (Act.) DGB129 and DGB70 MR1T cell clones (analysis performed by PageRank algorithm).

The following figure supplement is available for figure 6:

**Figure supplement 1.** FACS analysis of resting and activated DGB129 and DGB70 MR1T cells used for transcriptome studies.

---

*2007*). Both resting MR1T and MAIT cell clones displayed high levels of CXCR3 (with the exception of the DGA4 clone only; *Figure 7B*), However, we observed divergent expression patterns of CCR4 and CCR6 (*Figure 7B*), which further suggested that MR1T cells are heterogeneous and different from MAIT cells.

Taken together, these data indicated that the MR1-reactive clones tested here are phenotypically and functionally heterogeneous, thus suggesting that MR1T cells include multiple subsets with diverse recirculation patterns and tissue homing capacity and likely different specialized roles in human immunity.

## MR1T cells recognize cells constitutively expressing low surface MR1 and show diverse T helper-like functions

The MR1 expression level on the surface of APCs is physiologically regulated to be low to undetectable in the absence of microbial ligands (*Huang et al., 2008*; *McWilliam et al., 2016*). We therefore investigated whether monocyte-derived DCs (Mo-DCs) and other types of cells not-overexpressing MR1, but constitutively displaying low surface MR1 levels (*Figure 1A*, *Figure 8—figure supplement*

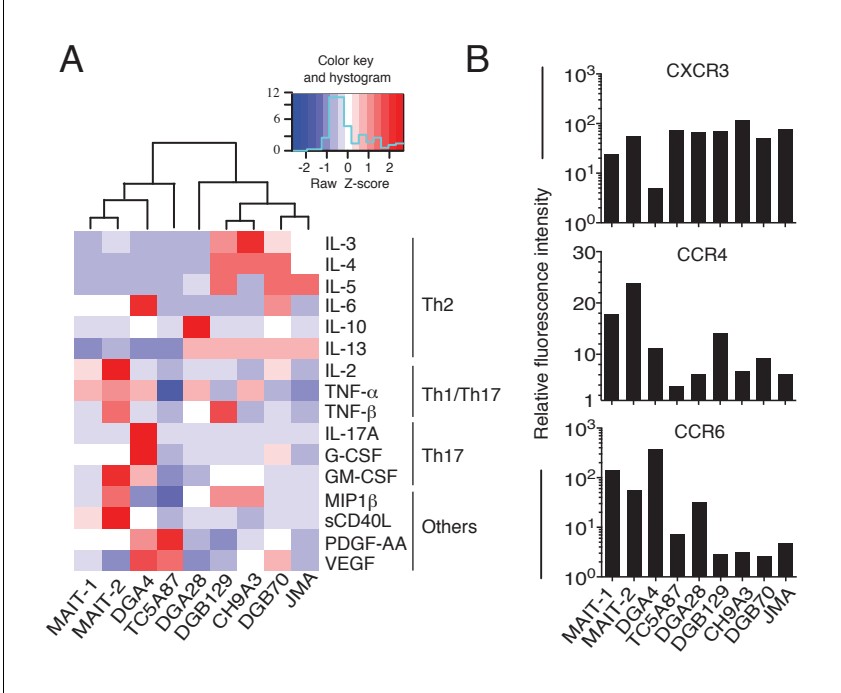

**Figure 7.** Functional heterogeneity of MR1T cell clones. (**A**) Heat-map of cytokine expression by seven different MR1T cell clones when stimulated by MR1-expressing A375 cells. Also shown are the cytokine profiles of control MAIT cell clones MRC25 (MAIT-1) and SMC3 (MAIT-2) following activation by A375-MR1 cells pulsed with *E. coli* lysate. Cytokines were assessed in the supernatants of duplicate cultures. The mean value for each cytokine was used to generate the heat-map. Cluster analysis was performed using Pearson correlation. Graphs displaying the amounts of individual cytokines released by the different clones are shown in *Figure 7—figure supplement 1*. (**B**) Flow cytometry analysis of CXCR3, CCR4 and CCR6 surface expression by resting MR1T cell clones and 2 MAIT control clones (MRC25, MAIT-1 and SMC3, MAIT-2). Graphs show the relative fluorescence intensity calculated by dividing the median fluorescence intensity (MFI) of specific mAb staining by the MFI of the corresponding isotype control. Data are representative of two independent experiments.

The following figure supplement is available for figure 7:

**Figure supplement 1.** Cytokines released by antigen-stimulated MR1T and MAIT cell clones.

---

*1A,C* and data not shown), were able to stimulate MAIT and MR1T cells. All these cell types, supported MAIT cell activation in the presence of microbial antigens and in an MR1-dependent manner (*Figure 8A*). The same cells also induced sterile MR1T cell activation to various extents. Mo-DC and THP-1 cells were recognized by the majority of the tested MR1T cell clones, followed by the Huh7 hepatoma cells, the LS 174T goblet-like cells and the HCT116 colon carcinoma cells (*Figure 8B*). Importantly, all responses were blocked by anti-MR1 mAbs.

Having found that MR1 T cells can recognize cells constitutively expressing physiological surface levels of MR1, we next investigated whether they can also modulate the functions of these target cells. First we focused on Mo-DCs, which stimulated several tested MR1T cell clones. The representative DGB129 MR1T cell clone reacted to Mo-DC from different donors (*Figure 8—figure supplement 1B*) and induced up-regulation of the maturation markers CD83 and CD86 on these target cells (*Figure 8C*). Remarkably, Mo-DC maturation promoted by DGB129 cells was fully inhibited by anti-MR1 mAbs, thus confirming that recognition of MR1 was required for this T helper-like function.

As we observed that three MR1T cell clones reacted also to LS 174T goblet-like cells, we next investigated the outcome of this interaction. LS 174 T cells are used as a model to study mucin gene expression regulation in intestinal epithelial cells (*van Klinken, 1996*) and express low surface levels of MR1 (*Figure 8—figure supplement 1C*). The selected JMA MR1T cell clone reacted to LS 174T by secreting in MR1-dependent fashion IL-8 and IL-13 (*Figure 8D,E*), two cytokines involved in the

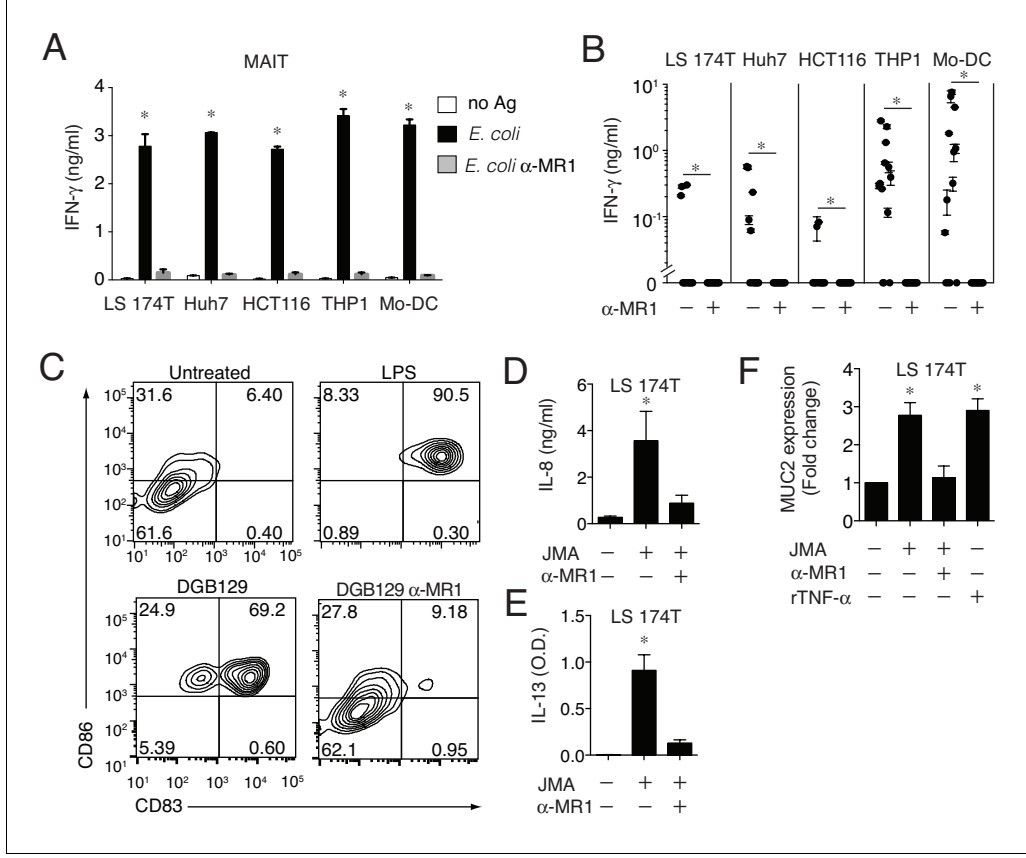

**Figure 8.** MR1T cells recognize cells constitutively expressing low surface MR1 and show diverse T helper-like functions. (**A**) Recognition of four human cells lines expressing constitutive surface levels of MR1 and Mo-DCs by the representative SMC3 MAIT cell clone in the absence (no Ag) or presence of *E. coli* lysate (*E. coli*) with or without anti-MR1 blocking mAbs (α-MR1). (**B**) Recognition of the same cell types as in A by thirteen MR1T cell clones with or without anti-MR1 mAbs (α-MR1). Graphs show IFN-γ release (mean ±SD of duplicate cultures). (**C**) Flow cytometry analysis of co-stimulatory molecules CD83 and CD86 on Mo-DCs after co-culture with DGB129 MR1T cells with or without anti-MR1 mAbs (α-MR1). A control group consisting of Mo-DCs stimulated with LPS (10 ng/ml) in the absence of T cells is also shown. Numbers indicate percentages of cells in each quadrant. (**D,E**) Stimulation of JMA MR1T cells by LS 174T intestinal epithelial cells with or without anti-MR1 mAbs (α-MR1). Columns show (**D**) IL-8 (ng/ml) and (**E**) IL-13 (optical density, O.D.) release. (**F**) Q-PCR analysis of *MUC2* gene expression in LS 174 T cells cultured alone or with JMA MR1T cells in the presence or absence of anti-MR1 mAbs (α-MR1). As control, LS 174 T cells were stimulated with recombinant TNF-α (rTNF-α, 10 ng/ml) in the absence of MR1T cells. All data are expressed as mean ± SD of triplicate cultures. Results are representative of at least three independent experiments. *p<0.05 (Unpaired Student's *t*-test).

The following figure supplement is available for figure 8:

**Figure supplement 1.** MR1 surface expression by monocyte-derived dendritic cells and LS 174 T cells.

modulation of human intestinal epithelial cell functions (*Iwashita et al., 2003*; *Sturm et al., 2005*). Interestingly, JMA recognition of LS 174 T cells also promoted their reciprocal activation resulting in increased transcription of mucin 2 gene (*MUC2*), which was prevented by anti-MR1 blocking antibodies (*Figure 8F*). In control experiments, similar *MUC2* expression upregulation was observed in LS 174 T cells treated with TNF-α (*Iwashita et al., 2003*) in the absence of MR1T cells (*Figure 8F*). These results unraveled the capacity of some MR1T cells to modulate *MUC2* expression in intestinal epithelial cells, and therefore suggested their possible contribution to intestinal epithelial barrier homeostasis.

Taken together, these data identify MR1T cells as a novel population of human MR1-restricted T lymphocytes that may mediate diverse immunological functions. These findings indicate that the repertoire and reactivity of MR1-restricted T cells in healthy individuals is far broader than was previously appreciated.

## Discussion

Here we report that functionally diverse human T cells respond to MR1 in the absence of microbial antigens, thus extending the role of MR1 beyond the presentation of microbial riboflavin precursors to MAIT cells. While the structure of MR1 resembles that of MHC class I and related proteins, this molecule also displays various unique properties. A key distinguishing feature is the presence of a bulky antigen-binding pocket that binds small molecules via either hydrophilic or hydrophobic interactions. The total volume of the MR1 antigen-binding pocket is far larger than any of the MR1 ligands reported to date, suggesting capacity to bind a wider repertoire of antigens than that known so far. A direct consequence of this is the likely existence of other T cell populations that recognize MR1-presented antigens distinct from the known riboflavin precursors or folate-derivatives. Our study indicates the existence of an unpredicted population of MR1-restricted T cells (designated MR1T cells) outside of the MAIT cell compartment. Human MR1T cell clones displayed differential recognition of a variety of target cells expressing constitutive levels of MR1. The different recognition of target cells might reflect different antigen presentation capacity, although they equally presented microbial antigens to MAIT cells, or the presence of different antigens. Indeed, preliminary antigen purification from a representative target cell line identified two diverse fractions that each stimulated only one of two MR1T cell clones, both derived from the same donor. Although the nature of these molecules remains to be determined, they do not derive from RPMI 1640 culture medium and are present in tumor cells grown in vivo. Furthermore, initial characterization of the molecules showed that they formed stable complexes with plastic-bound MR1 without forming a Schiff base and activated specific MR1T cells without need of APC processing.

Using as APCs A375 cells expressing high levels of MR1, we estimated that MR1T cells represent between 1:2500 and 1:5000 of total circulating T cells. These frequency estimates could not be determined by direct ex vivo identification of MR1T cells due to lack of specific markers, and were obtained using two approaches. In the first, frequencies were calculated after overnight stimulation and subtracting the frequency of T cells responding to MR1-negative APC to the frequency of cells responding to MR1-positive APC. In another set of experiments the overnight-stimulated T cells were immediately cloned without previous in vitro expansion and individual MR1T cell clones were identified with functional assays. These observations indicated that MR1T cell are readily detectable in the blood of healthy individuals with a size that resembles that of MHC-restricted memory T cells recognizing pathogen-derived peptides in pathogen-exposed individuals (*Bacher and Scheffold, 2013*; *Lucas et al., 2004*; *Su et al., 2013*; *Yu et al., 2015*). The calculated range of MR1T cell frequency could however be underestimated as A375 cells utilized for the frequency studies might lack some MR1T cell antigens and therefore fail to stimulate a fraction of MR1T cells. Collectively, our data suggest that MR1T cells are not rare T lymphocytes in human blood.

In some respects, MR1T cells resemble another population of autoreactive T cells restricted to non-polymorphic antigen presenting molecules and present at high frequency in human blood. These T cells react to self-lipids presented by CD1 molecules (*de Jong, 2010*; *de Lalla, 2011*; *Dellabona et al., 2015*). In analogy to MR1T cells, CD1-self-reactive T cells display a polyclonal TCRα and β usage, and are functionally heterogeneous (*de Jong, 2010*; *de Lalla, 2011*; *Dellabona et al., 2015*). The physiological role of self-reactive T cells restricted by CD1 and MR1 molecules is not well characterized. It is tempting to speculate that these cells might be activated upon metabolic changes in target cells induced by environmental signals, infection or cell transformation.

It is currently unclear whether MR1T cells can be classified into distinct subgroups based on phenotype, gene expression and function. Since MR1T cells exhibit non-invariant TCRs with diverse antigen specificities, express a wide range of different surface markers and release multiple soluble effector molecules, it will be challenging to identify these cells using MR1 tetramer staining combined with conventional phenotypic analyses. Interestingly, some activated MR1T cell clones also displayed atypical functions for T cells, including the release of growth factors VEGF and PDGF, which

support mesenchymal cell proliferation, vasculogenesis, and angiogenesis (*Carmeliet, 2003*), suggesting a potential role in tissue remodeling processes.

The functional variety of MR1T cells was also accompanied by differential expression of selected chemokine receptors, suggesting that this population may exert multiple functions at various sites throughout the body. By comparing the transcriptomes of two select MR1T cell clones, we observed that certain transcription factors were expressed in common by both cell types, whereas others were only expressed in a clone-specific fashion. The DGB70 clone was distinguished by expression of the regulator of T cell polarization and proliferation factor *EGR-1* (*Shin et al., 2009*; *Thiel and Cibelli, 2002*), the mediator of lymphocyte functional differentiation *JUN-B* (*Li et al., 1999*), and the *SREBF1*, a member of the *SREBP* gene family that regulates sterol synthesis, coordinates tissue-specific gene expression and controls T cell metabolism (*Kidani et al., 2013*; *Shao and Espenshade, 2012*). In contrast, transcription factors unique to the DGB129 clone were the proliferation and apoptosis regulator *MYC* (*Kress et al., 2015*), involved in metabolic programming of effector T cells (*Hough et al., 2015*), and the cAMP responsive element binding factor *CREM,* which modulates T cell effector functions and cytokine gene expression (*Rauen et al., 2013*). DGB129 cells also uniquely upregulated the *HSP90AB1* heat shock protein gene, which mediates quality control of cytoplasmic proteins (*Taipale et al., 2010*) and is involved in the regulation of diverse cell surface receptor expression in T and NK cells (*Bae et al., 2013*). These clear distinctions between the transcriptional profiles of the two MR1T cell clones indicate the use of different strategies for metabolic programming, cell cycle control and functional polarization and suggest specific roles for these cells. As sorted cells expressed equivalent levels of CD25 and CD137 after Ag stimulation (see *Figure 6—figure supplement 1*), and released comparable amounts of IFN-γ (not shown) we excluded a different strength of antigen stimulation as possible explanation of the diverse transcriptional profiles observed. Given that the two clones were isolated from the same donor, maintained in identical culture conditions and stimulated with the same type of APC, these divergent gene signatures could reflect unique priming events (*Boltjes and van Wijk, 2014*).

A key finding of our study is the reactivity of a large fraction of MR1T cell clones to diverse types of target cells expressing constitutive low MR1 surface levels. These results indicated the capacity of MR1T cells to be activated in physiologic conditions, when limiting amounts of MR1-antigen complexes are available on the target cell surface. As consequence, these data raised the issue of what influence MR1T cells could exert on these target cells. Several MR1T clones tested recognized monocyte-derived DCs, thus suggesting that MR1-restricted reactivity to these APCs is quite frequent among MR1T cells. In addition, experiments performed with a representative MR1T cell clone revealed its capacity to promote monocyte-derived DCs maturation. These data suggest that MR1T cells may modulate DC functions as it has already been reported for other T lymphocyte populations including TCR γδ (*Devilder et al., 2006*) and CD1-restricted T cells (*Bendelac et al., 2007*; *Vincent et al., 2002*). Some MR1T cell clones were also stimulated by human intestinal epithelial LS 174 T cells. In co-culture experiments, we observed reciprocal MR1-dependent activation of both MR1T and LS 174 T cells, with the first releasing IFN-γ, IL-8 and IL13, and the latter up-regulating mucin 2 gene expression. These data suggested that some MR1T cells might influence epithelial cell function at mucosal sites, perhaps promoting innate defense and/or inflammation.

In conclusion, we report that in addition to microbial metabolite-sensitive MAIT cells, human blood contains a novel population of MR1-restricted T cells that can recognize different antigens present in distinct target cells and exhibits a variety of effector functions. Based on the diversity of effector molecules they release following antigen stimulation, MR1T cells might drive inflammatory responses, support B cell function, mediate DC licensing, promote tissue remodeling, and contribute to mucosal homeostasis by enhancing innate defenses at the epithelial barrier. Future studies are therefore likely to uncover the roles of MR1T cells in human diseases.

## Materials and methods

### Cells

The following cell lines were obtained from American Type Culture Collection: A375 (human melanoma), THP-1 (myelomonocytic leukemia), J.RT3-T3.5 (TCRβ-deficient T cell leukemia), HEK 293 (human embryonic kidney), CCRF-SB (acute B cell lymphoblastic leukemia), Huh7 (human hepatoma),

HCT116 (human colorectal carcinoma), LS 174T (goblet-like cells from colon adenocarcinoma), and EMT6 (mouse breast carcinoma). SKW-3 cells (TCRα- and β-deficient human T cell leukemia) were obtained from the Leibniz-Institute DSMZ-German Collection of Microorganisms and Cell Cultures. All the used cells were routinely tested for mycoplasma contamination and were negative. None of the cell lines used in this study is present in the database of commonly misidentified cell lines. Cells lines were not authenticated. Two representative MAIT clones (MRC25 and SMC3) were used in this study generated from blood of two different healthy donors and maintained in culture as previously described (*Lepore et al., 2014*). MR1T cells were isolated from the peripheral blood of healthy individuals. Briefly, T cells purified by negative selection were stimulated with irradiated (80 Gray) A375-MR1 cells (ratio 2:1) once a week for three weeks. Human rIL-2 (5 U/ml; Hoffmann-La Roche) was added at day +2 and+5 after each stimulation. Twelve days after the last stimulation cells were washed and co-cultured overnight with A375-MR1 cells (ratio 2:1). CD3$^+$CD69$^+$CD137$^{high}$ cells were then sorted and cloned by limiting dilution in the presence of PHA (1 μg/ml, Wellcome Research Laboratories), human rIL-2 (100 U/ml, Hoffmann-La Roche) and irradiated PBMC (5 × 10$^5$ cells /ml). In other experiments, MR1 T cells clones were generated using the same protocol from sorted CD3$^+$CD69$^+$CD137$^{high}$ upon a single overnight stimulation with A375-MR1 cells (ratio 2:1). T cell clones were periodically re-stimulated following the same protocol (*Lepore et al., 2014*). Monocytes and B cells were purified (>90% purity) from PBMCs of healthy donors using EasySep Human CD14 and CD19 positive selection kits (Stemcell Technologies) according to the manufacturer instructions. Mo-DCs were differentiated from purified CD14$^+$monocytes by culture in the presence of GM-CSF and IL-4 as previously described (*Lepore et al., 2014*).

## Generation of cells expressing MR1A gene covalently linked with β2m

A human MR1A cDNA construct linked to β2m via a flexible Gly-Ser linker was generated by PCR as previously described (*Lepore et al., 2014*). The K43A substitution in the MR1A cDNA was introduced into the fusion construct using the following primers: MR1K43A_f 5′-CTCGGCAGGCCGAGC-CACGGGC and MR1K43A_r 5′GCCCGTGGCTCGGCCTGCCGAG. Resulting WT and mutant constructs were cloned into a bidirectional lentiviral vector (LV) (*Lepore et al., 2014*), provided by Jürg Schwaller, Department of Biomedicine, University of Basel. HEK293 cells were transfected with individual LV-MR1A-β2m constructs together with the lentivirus packaging plasmids pMD2.G, pMDLg/pRRE and pRSV-REV (Addgene) using Metafectene Pro (Biontex) according to the manufacturer instructions. A375, THP-1, CHO and J558 cells were transduced by spin-infection with virus particle containing supernatant in presence of 8 μg/ml protamine sulfate. Surface expression of MR1 was assessed by flow cytometry and positive cells were FACS sorted.

## Soluble recombinant β2m-MR1-Fc fusion protein

β2m-MR1-Fc fusion construct was obtained using human MR1A-β2m construct described above as template. DNA complementary to β2m-MR1A gene was amplified by PCR using primers: β2mXhoI_f 5′- CTCGAGATGTCTCGCTCCGTGGCCTTA and MR1-IgG1_r 5′-GTGTGAGTTTTGTCGCTAGCC TGGGGGGACCTG, thus excluding MR1 trans-membrane and intracellular domains. The DNA complementary to the hinge region and CH2-CH3 domains of human IgG1 heavy chain was generated using the following primers: NheI-hinge-f 5′-CAGGTCCCCCCAGGCTAGCGACAAAACTCACAC and IgG1NotI_r 5′-GCGGCCGCTCATTTACCCGGAGACAGGGAGA from pFUSE-hIgG1-Fc1 (InvivoGen). The β2m-MR1A and IgG1 PCR products were joined together using two-step splicing with overlap extension PCR and the resulting construct subcloned into the XhoI/NotI sites of the BCMGSNeo expression vector. CHO-K1 cells were transfected with the final construct using Metafectene Pro (Biontex), cloned by limiting dilutions and screened by ELISA for the production of β2m-MR1-Fc fusion protein. Selected clones, adapted to EX-CELL ACF CHO serum-free medium (Sigma), were used for protein production and β2m-MR1-Fc was purified using Protein-A-Sepharose (Thermo Fisher Scientific) according to the manufacturer instructions. Protein purity was verified by SDS-PAGE and Western Blot. Protein integrity was assessed by ELISA, using anti-β2M mAb (HB28) as capture and the conformation-dependent anti-MR1 mAb 25.6 as reveling (*Kjer-Nielsen et al., 2012*).

## Flow cytometry and antibodies

Cell surface labeling was performed using standard protocols. Intracellular labeling was performed using the True-Nuclear Transcription Factor Buffer Set according to the manufacturers' instructions. The following anti-human mAbs were obtained from Biolegend: CD4-APC (OKT4), CD8α-PE (TuGh4), CD161-Alexa Fluor 647 (HP-3G10), CD69-PE (FN50), CD3-PE/Cy7, Brilliant Violet-711, or Alexa-700 (UCHT1), CD137-biotin (n4b4-1), CXCR3-Brilliant Violet 421 (G025H7), CD83-biotin (HB15e) and TRAV1-2- PE (10C3). CD86-FITC (2331), CCR4-PECy7 (1G1) and CCR6-PE (11A9) mAbs were from BD Pharmingen. All these mAbs were used at 5 µg/ml. Biotinylated mAbs were revealed with streptavidin-PE, -Alexa Fluor 488, or -Brilliant violet 421 (2 µg/ml, Biolegend). The MR1-specific mAb clone 26.5 (mouse IgG2a) was provided by Ted Hansen, Marina Cella and Marco Colonna, Washington University School of Medicine, St. Louis (MO) (*Lepore et al., 2014*). Unlabeled MR1-specific mAbs were revealed with goat anti-mouse IgG2a-PE (2 µg/ml, Southern Biotech). Samples were acquired on LSR Fortessa flow cytometer (Becton Dickinson). Cell sorting experiments were performed using an Influx instrument (Becton Dickinson). Dead cells and doublets were excluded on the basis of forward scatter area and width, side scatter, and DAPI staining. All data were analyzed using FlowJo software (TreeStar).

## TCR gene analysis of MR1T cell clones

TCRα and β gene expression by MR1T cell clones was assessed either by RT-PCR using total cDNA and specific primers, or by flow cytometry using the IOTest Beta Mark TCR Vβ Repertoire Kit (Beckman Coulter) according to the manufacturers' instructions. For RT-PCR, RNA was prepared using the NucleoSpin RNA II Kit (Macherey Nagel) and cDNA was synthesized using Superscript III reverse transcriptase (Invitrogen). TCRα and β cDNAs were amplified using sets of Vα and Vβ primers as directed by the manufacturer (TCR typing amplimer kit, Clontech). Functional transcripts were identified by sequencing and then analyzed using the ImMunoGeneTics information system (http://www.imgt.org). The cDNA-sequencing data set has been deposited in the GenBank repository under accession numbers MF085360-MF085372.

## Fractionation of cell and whole tumor lysates

Total cell lysates were generated from a single pellet of $2.5 \times 10^9$ THP-1 cells via disruption in water with mild sonication. The sonicated material was then centrifuged (15,000 g for 15 min at 4°C) and the supernatant collected (S1). Next, the pellet was re-suspended in methanol, sonicated, centrifuged (15,000 g for 15 min at 4°C), and the supernatant obtained (S2) was pooled with the S1 supernatant. The final concentration of methanol was 10%. The total cell extract (pool of S1 and S2) was then loaded onto a C18 Sep-Pak cartridge (Waters Corporation) and the unbound material was collected and dried (fraction E-FT). Bound material was eluted in batch with 75% (fraction E1) and 100% methanol (fraction E2), collected and dried. The E-FT material was re-suspended in acetonitrile/water (9:1 vol/vol) and loaded onto a $NH_2$ Sep-Pak cartridge (Waters Corporation). Unbound material (fraction N-FT) and 4 additional fractions (N1-N4) were eluted with increasing quantities of water. Fraction N1 was eluted with 35% $H_2O$, fraction N2 with 60% $H_2O$, fraction N3 with 100% $H_2O$, and fraction N4 with 100% $H_2O$ and 50 mM ammonium acetate (pH 7.0). Fractions N1-N4 were then dried and stored at −70°C until use. All collected dry-frozen fractions were then thawed and re-suspended in $H_2O$ 20% methanol (fractions E1, E2 and N-FT) or 100% $H_2O$ (all other fractions) prior to being tested in T cell activation assays.

Mouse EMT6 breast tumors were provided by A. Zippelius and were prepared as described (*Zippelius et al., 2015*). Freshly excised tumors were extensively washed in saline, weighted and 4 g masses were homogenized in 7 ml of HPLC-grade water using a Dounce tissue grinder. Tumor homogenate underwent two freeze-thaw cycles, centrifuged (3250 g) for 10 min at 4°C, and supernatant was collected and stored at −70°C. The pellet was extracted a second time with 2 ml of HPLC-grade water, centrifuged (5100 g) for 10 min at 4°C and the supernatant was collected and stored at −70°C. The pellet was further extracted with 9 ml of HPLC-grade methanol for 5 min at room temperature by vortexing, centrifuged (5100 g) for 10 min at 4°C, and supernatant collected. The three supernatants were pooled, dried, and resuspended in water:methanol (10:1). Material was fractionated using C18 and $NH_2$ Sep-Pak cartridges as above.

## T cell activation assays

MR1-restricted T cells ($5 \times 10^4$/well unless otherwise indicated) were co-cultured with indicated target cells ($5 \times 10^4$/well) in 200 µl total volume in duplicates or triplicates. T cells were cultured with indicated APCs for 24 hr. In some experiments, anti-MR1 mAbs (clone 26.5) or mouse IgG2a isotype control mAbs (both at 30 µg/ml) were added and incubated for 30 min prior to the addition of T cells. In some experiments A375-MR1 and THP1-MR1 cells were cultured four days in PBS supplemented with 5% human AB serum. Cells were washed every day and then used to stimulate T cell clones upon assessment of target cell viability by trypan blue staining. In such type of experiments, T cell activation assays were also performed in PBS supplemented with 5% human AB serum. *E. coli* lysate was prepared from the DH5α strain (Invitrogen) grown in LB medium and collected during exponential growth. Bacterial cells were washed twice in PBS and then lysed by sonication. After centrifugation (15,000 g for 15 min), the supernatant was collected, dried, and stored at −70℃. APCs were pulsed for 4 hr with *E. coli* lysate equivalent to $10^8$ CFU/ml (unless otherwise indicated) before addition of T cells. In some experiments, APCs were pre-incubated with 6-FP or Ac-6-FP (Schircks Laboratories) or pulsed with 6,7-dimethyl-8-D-ribityllumazine (RL-6,7-diMe; 150 µM; Otava Chemicals) for 4 hr before co-culture with T cells. In control experiments with TCR γδ cells, the APCs were first treated for 6 hr with zoledronate (10 µg/ml, Sigma) prior to T cell addition. Activation experiments with plate-bound recombinant human β2m-MR1-Fc were performed by coating β2m-MR1-Fc onto 96 well plates (4 µg/ml) and loading with cartridge-purified cell lysates for 4 hr at 37℃ before washing twice and adding T cells. Supernatants were collected after 24 hr and IFN-γ or GM-CSF were assessed by ELISA. Multiple cytokines and chemokines in cell culture supernatants were analyzed using the Milliplex MAP human cytokine/chemokine magnetic bead panel – Premixed 41-plex (HCYTMAG-60K-PX41; Merck Millipore) according to the manufacturer's instructions. Samples were acquired on a Flexmap 3D system (Merck Millipore) and Milliplex analyst software was used to determine mean fluorescence intensity and analyte concentration.

## Modulation of DC and epithelial cell functions

Mo-DCs were co-cultured with MR1T cell clones at 1:1 ratio. When indicated Mo-DCs were pre-incubated with anti-MR1 mAbs (clone 26.5, 30 µg/ml) for 30 min prior addition of T cells. Supernatants were harvested after 48 hr and tested for IFN-γ release by ELISA to assess T cell activation. Cells were collected at the same time and analyzed by FACS for CD83 and CD86 expression. T cells were excluded by the analysis using anti-CD3 mAbs. LS 174 T cells were co-cultured with MR1T cell clones at 1:1 ratio in the presence or absence of anti-MR1 (clone 26.5) or mouse IgG2a isotype control mAbs (both at 30 µg/ml, incubated with APCs 30 min prior addition of T cells). MR1T cell activation was evaluated by measuring IFN-γ, IL-8 (R&D) and IL-13 (Pharmingen) in the supernatants by ELISA. Q-PCR for *MUC2* gene expression by LS 174 T cells was performed in a 20 µl reaction volume containing 0.5 µM of each primer and 2.5 µl of cDNA in Power SYBRgreen MasterMix (Applied Biosystems). β-actin was used as reference gene. All the reactions were carried out in triplicate. The following method was run using an ABI 7500 Fast Real-Time PCR System (Applied Biosystem): initial incubation for 20 s at 50℃, incubation for 10 min at 95℃, 40 cycles of 15 s at 95℃ and 1 min at 60℃. The PCR primers used were as follows: β-actin_f 5′-GCCACCCGCGAGAAGATGA, β-actin_r 5′-CATCACGATGCCAGTGGTA; MUC2_f 5′-ACCCGCACTATGTCACCTTC, MUC2_r 5′-GGGATCGCAGTGGTAGTTGT. Changes in gene expression were quantified using the ΔΔCt method.

## TCR gene transfer

TCRα and β functional cDNA from the MAIT cell clone MRC25 were cloned into the *XhoI*/*NotI* sites of the BCMGSNeo expression vectors and the resulting constructs were used to co-transfect J.RT3-T3.5 cells by electroporation. Transfectants expressing TRAV1-2 and CD3 were FACS sorted. The TCRα and β functional cDNA from MR1T clones were cloned into the *XmaI*/*SalI* sites of a modified version of the p118 lentiviral expression vector (*Amendola et al., 2005*; *Lepore et al., 2014*). SKW-3 cells were transduced with virus particle-containing supernatant generated as described above. Cells were FACS sorted based on CD3 expression.

## Next-generation sequencing, transcriptome and gene centrality analyses

Comparative transcriptome analysis between resting and A375-MR1 stimulated MR1T cell clones was performed by RNA-sequencing. T cells were stimulated with A375-MR1 cells and after 20 hr DAPI$^-$CD3$^+$CD25$^+$CD137$^+$ cells were sorted, counted, and immediately frozen. Total RNA was extracted using Arcturus PicoPure RNA Isolation kit (Applied Biosystems) according to the manufacturer's protocol. RNA quality, assessed using the Agilent Bioanalyzer, exhibited RNA integrity number (RIN) $\geq$6.9. Next, cDNA libraries were prepared using 200 ng total RNA and a 2 µl volume of a 1:500 dilution of ERCC RNA Spike in Controls (Ambion). Samples were subjected to cDNA library synthesis using the TruSeq Stranded mRNA Library Preparation Kit (Illumina) according to the manufacturer's protocol with minor modifications; 13 PCR cycles used, and two additional rounds of purification with Agencourt Ampure XP SPRI beads (Beckman Courter) to remove double-stranded cDNA >600 bp in size. The length distribution of the cDNA libraries was monitored using DNA 1,000 kits with the Agilent Bioanalyzer. The samples were then subjected to an indexed PE sequencing run of 2 × 51 cycles on an Illumina HiSeq 2000.

Raw RNA-seq reads in fastq format were aligned to the hg19 genome assembly using STAR aligner (*Dobin et al., 2013*). Gene annotations were derived from GENCODE version (*Harrow et al., 2012*) and counts of reads mapping over gene features were obtained using the featureCount method of the R/bioconductor subread packages (*Liao et al., 2014*). EdgeR (*Robinson et al., 2010*) was used to generate a differentially expressed gene (DEG) list. From the DEGs, a global network of transcription factors and their target genes was created using the Encode chip-seq data as described (*Narang et al., 2015*). Using the list of DEGs (FDR < 0.05, log$_2$ fold-change >2), a subnetwork of the global transcription factors-gene network was created and centrality scores for nodes were generated by both PageRank and betweeness centrality algorithms using NetworkX library in Python. The network was visualized using Cytoscape software and betweeness centrality score were used to define the radius of the nodes (circles in the network). The RNA-sequencing data set has been deposited in the Gene Expression Omnibus repository under accession code GSE81063.

## Statistics

For cytokine secretion assays and Q-PCR data were analyzed using Unpaired Student's *t*-test (Prism 6, GraphPad software).

## Acknowledgements

We thank L Angman and C Furtwängler for TCR cloning and gene-transfer experiments, E Traunecker for cell sorting, S Sansano for MR1 purification, T Rutishauer for help with figure preparation, A Zippelius for EMT6 mouse tumors, J Schwaller, G Spagnoli and V Governa for providing reagents (all from University of Basel), D Trono for plasmid availability through Addgene, T Hansen, M Cella, M Colonna (University of Washington) for MR1-specific mAbs. We thank O Lantz (Institut Curie, Paris) for helpful discussions and N McCarthy of Insight Editing London for reviewing the manuscript.

## Additional information

### Funding

| Funder | Grant reference number | Author |
|---|---|---|
| Schweizerischer Nationalfonds zur Förderung der Wissenschaftlichen Forschung | 310030-149571 | Gennaro De Libero |
| European Commission | 643381 | Gennaro De Libero |
| Science and Engineering Research Council | 1121480006 | Gennaro De Libero |
| Universität Basel | Core Funding | Gennaro De Libero |

| | | |
|---|---|---|
| Agency for Science, Technology and Research | 1201826277 | Gennaro De Libero |
| DBM, University Hospital Basel | Core Funds | Gennaro De Libero |
| Horizon 2020 Framework Programme | "TBVAC 2020" 643381 | Gennaro De Libero |
| Agency for Science, Technology and Research | SIgN, Core Funds | Lucia Mori Gennaro De Libero |
| Universität Basel | Förderbeiträge für exzellentejunge Forschende | Marco Lepore |

The funders had no role in study design, data collection and interpretation, or the decision to submit the work for publication.

### Author contributions

ML, Conceptualization, Data curation, Formal analysis, Validation, Investigation, Visualization, Methodology, Writing—original draft, Writing—review and editing; AK, Validation, Investigation, Methodology; SC, BP, MS, FZ, Formal analysis, Investigation, Methodology; PK, Software, Formal analysis, Investigation, Visualization, Methodology; VN, Resources, Software, Methodology; MP, Software, Formal analysis, Supervision, Methodology; LM, Conceptualization, Resources, Data curation, Supervision, Validation, Investigation, Visualization, Writing—original draft, Writing—review and editing; GDL, Conceptualization, Resources, Supervision, Funding acquisition, Investigation, Methodology, Writing—original draft, Project administration, Writing—review and editing

### Author ORCIDs

Marco Lepore, http://orcid.org/0000-0003-1353-8224
Lucia Mori, http://orcid.org/0000-0002-5522-4648
Gennaro De Libero, http://orcid.org/0000-0003-0853-7868

### Ethics

Human subjects: Venous blood was taken from healthy donors after informed consent obtained at the time of blood collection under approval of the "Ethikkommision Nordwest und Zentralschweiz/ EKNZ (139/13).

## Additional files

### Supplementary files

• Supplementary file 1. Genes modulated in activated *vs.* resting MR1T cell clones.

### Major datasets

The following dataset was generated:

| Author(s) | Year | Dataset title | Dataset URL | Database, license, and accessibility information |
|---|---|---|---|---|
| Lepore M, De-Libero G, Mori L | 2016 | Transcriptome analysis of MR1 reactive T cells | http://www.ncbi.nlm.nih.gov/geo/query/acc.cgi?acc= GSE81063 | Publicly available at the NCBI Gene Expression Omnibus (accession no: GSE81063) |

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
