## [Decision Letter]

Thank you for submitting your article "Functionally diverse human T cells recognize non-microbial antigens presented by MR1" for consideration by *eLife*. Your article has been favorably evaluated by Tadatsugu Taniguchi (Senior Editor) and three reviewers, one of whom is a member of our Board of Reviewing Editors. The reviewers have opted to remain anonymous.

Summary:

The paper by Lepore et al. identifies and determines the specificity of MR1-restricted T cells in human peripheral blood. In contrast to current concepts of MR1-restricted T cells, termed mucosal associated invariant T cells that are also MR1-restricted, the MR1-reactive T cells studied here appear to be specific for non-microbial ligands. While the ligands have not been identified at the molecular level, the data presented appear to be quite convincing for this conclusion. As such, the paper provides an advance for understanding of MR1-restricted T cells because they now appear to be specific for non-microbial antigens.

The manuscript was evaluated by a member of the Board of Reviewing Editors and external referees. After a thorough online discussion, they came to the consensus that the paper is interesting but the authors need to address several major points as outlined below. As is often the case, specific critiques become more apparent with discussion. In any event, the full reviews of the referees are shown for completeness but the authors should be encouraged to submit a revised manuscript that addresses all of the major concerns in the consensus review.

Essential revisions:

1) Address the level of MR1 on the APCs that you used. It seems that the A375-MR1 cell expresses very high levels that may explain why others have not seen MR1-restricted stimulation previously. What happens when you use an APC with lower MR1 expression?

2) Address why CD137 was used as an activation marker. This seems to be atypical as with CD69 expression only, the MR1 specific response was undetectable (Figure 3).

3) Describe the epitope recognized by the anti-MR1 blocking antibody. Verify its monospecificity.

4) Address the limitations of the transcriptional and cytokine profiles. They appear to have been done on clones in culture for some time.

5) There are concerns about the accuracy of the precursor frequency estimates. Caveats should be more fully discussed. Also discuss more fully the significance of the estimates.

6) We expect that the TCR sequences will be submitted if the manuscript is published.

[Editors' note: further revisions were requested prior to acceptance, as described below.]

Thank you for resubmitting your work entitled "Functionally diverse human T cells recognize non-microbial antigens presented by MR1" for further consideration at eLife*eLife*. Your revised article has been favorably evaluated by Tadatsugu Taniguchi (Senior Eeditor), a Reviewing Eeditor, and one reviewer.

We apologize for the delay in reaching a decision, which was largely due to a tardy reviewer.

The manuscript has been improved but there is one quite significant issue that must be addressed before acceptance, as outlined below:

Although the authors have responded to most of the reviewers' comments, the reviewers and Editors are particularly concerned that the sequences of the TCR are being withheld because of a patent application. The authors' identification of the TCRs is based on the sequences, and those data are the primary data on which this conclusion is drawn. Also, the statement that they will make sequences available "for collaboration" is unacceptable. The sequences must be available on publication and deposited in a public database. These conditions are non-negotiable.

---

## [Author Response]

*Essential revisions:*

*1) Address the level of MR1 on the APCs that you used. It seems that the A375-MR1 cell expresses very high levels that may explain why others have not seen MR1-restricted stimulation previously. What happens when you use an APC with lower MR1 expression?*

We thank the reviewers for raising this important issue that we realized was not sufficiently clear in the submitted manuscript. We have changed the revised manuscript to clarify this issue (Results subsection “MR1-reactive T cells are readily detectable in healthy individuals” and Discussion, second paragraph).

It is very important to note that one of the major findings of our study is the reactivity of a large fraction of MR1T cell clones to APCs expressing constitutive, and indeed low, MR1 surface levels, including monocyte-derived dendritic cells, and some tumor cell lines (Figure 8). These results strongly indicated the capacity of MR1T cells to be activated in physiologic conditions, when limiting amounts of MR1-Ags complexes are exposed on the surface of APCs.

To test the hypothesis whether MR1-autoreactive non-MAIT T cells exist in peripheral blood, we used as stimulatory cells *ad hoc* APC that do not express classical MHC class I and CD1 molecules and instead express MR1. This approach was chosen, as there was no information in the literature about the existence of MR1-autoreactive T cells and there are no reagents allowing their direct identification, i.e.antigen-loaded tetramers.

in vitrostimulation of T cells with APC expressing high levels of antigen-presenting molecules or high doses of self-antigens is instrumental to reveal the existence of self- reactive T cells and this aspect has been recently discussed (1). Furthermore, the use of APC expressing large amounts of CD1 molecules was necessary to reveal self-reactive T cells restricted to CD1a and CD1c molecules (2-5).

In addition, several studies investigating MR1-restricted T cells utilized MR1- overexpressing APCs, which have been also important for the characterization of MAIT cell reactivity and the identification of the recognized antigens (6-8).

*2) Address why CD137 was used as an activation marker. This seems to be atypical as with CD69 expression only, the MR1 specific response was undetectable (Figure 3).*

The activation marker CD137 (4-1BB) is widely used to selectively identify antigen- specific T cells (9). CD69 also represents a very sensitive surface marker to detect early T cell activation, but if not analyzed in combination with other markers its high sensitivity may have some limitations (9). Firstly, some T cell subsets repetitively stimulated in vivo, i.e.NKT, MAIT cells and some TCR γδ cells, display constitutive high level of CD69 (10-12), which may mask specific stimulation of other less abundant T cell populations. Secondly, CD69 up-regulation by T cells can also occur in response to environmental stimuli, independently of T cell receptor-mediated antigen recognition (13, 14), and therefore it might give false positive results if used to study T cell antigen specificity (9). Thirdly, CD69 expression remains high on T cells repetitively stimulated in vitro(see new Figure 2 of the revised manuscript), thus becoming poorly informative in revealing antigen-dependent activation.

By contrast, CD137 expression is highly dependent on recent activation driven by T cell receptor engagement (15) and therefore is more restrictive than CD69 up-regulation to identify antigen-specific T cells (9, 16). In addition, CD137 expression allows selective detection of antigen-specific T cells belonging to CD8^+^, CD4^+^, and CD8^-^CD4^-^ subsets, differently from CD40L, which instead limits this capacity to CD4^+^ T cells. The reliability of CD137 has been successfully exploited also to identify rare T cells reactive to weak antigens, such as self- or tumor-associated antigens, both ex vivoand following in vitrostimulation (17). We selected the combination of CD69 and CD137 markers because of their complementary properties, the first being more sensitive the second more restrictive.

*3) Describe the epitope recognized by the anti-MR1 blocking antibody. Verify its monospecificity.*

The specificity of 26.5 anti-MR1 monoclonal antibodies utilized in this study was extensively characterized in two studies (18, 19), although the epitope recognized was not entirely mapped. This antibody cross-reacts with mouse, rat and bovine MR1, because of the high degree of MR1 conservation across these species, but not with any other known antigen-presenting molecules (18, 19). The 26.5 mAbs recognizes a conformation-depend epitope present only on properly loaded, and thus T cell stimulatory, MR1 molecules (18, 19). Importantly, specific mutations of residues present on the α-helices forming the MR1 groove prevent or impair the antibody binding to MR1 (18, 19). The authors interpreted the results as evidence that the binding occurs on the MR1 surface nearby the region interacting with MAIT TCR. This interpretation is in agreement with the mAb capacity of blocking MR1-mediated T cell activation (18, 19). The conformation-dependent specificity of 26.5 mAb has also been confirmed in ELISA using recombinant MR1 proteins by us (Materials and methods, subsection “Soluble recombinant β2m-MR1-Fc fusion protein”) and others (7). The 26.5 mAbs have been largely used to study specific MR1-dependent T cell stimulation. In some of our experiments, not included in the manuscript, we also utilized the 12.2 anti-MR1 mAbs (18, 19), which recognize a different conformation-dependent epitope of MR1. Also this mAb blocked both MAIT and MR1T cell activation similarly to 26.5 mAbs (our unpublished data).

*4) Address the limitations of the transcriptional and cytokine profiles. They appear to have been done on clones in culture for some time.*

The T cell clones used for the transcriptome studies are derived from the same donor, were kept in culture in identical conditions and stimulation was performed with the same antigen presenting cells on the same day. Despite these identical conditions, important differences were observed, indicating true differences among analyzed cells.

Great diversity in cytokine release following antigen stimulation was observed, confirming the divergent functions of the tested MR1T cell clones. A concern of the reviewers might be that in vitroexpansion could induce biases in the cytokines released by the clones. According to other studies (20), these biases affect the secretion of IFN-γ and not of other soluble factors. Indeed, T cell clones derived from ex vivosorted Th1, Th2 or Th17 cell populations, retained cytokine secretion patterns typical of the sorted functional subset and acquired the capacity of releasing IFN-γ upon in vitroexpansion (20). Accordingly, our MR1T clones, which were cultivated using conditions similar to those used in other studies (20), maintained differential patterns of cytokine release, and also secreted IFN-γ upon antigen stimulation.

The transcriptome analysis was performed by comparing T cell clones both in resting state and sorted following stimulation with the same APCs, according to equivalent expression levels of CD25 and CD137 (see Figure 6—figure supplement 1). In addition, we also measured the amounts of IFN-γ released by the T cell clones in the same experiments and it was comparable (not shown). Therefore, we are confident that a different strength of antigen stimulation might be excluded as possible explanation of the diverse transcriptional profiles observed.

The transcriptome analysis was performed using three biological replicates using three MR1T cell clones. We reasoned that with this approach it would have been possible to identify a common transcriptional signature unique to MR1T cells. Unfortunately, the data relative to the third clone did not pass the quality control and were excluded from the analysis. Importantly, despite highly stringent cut-off criteria (FDR<0.05, minimum log_2_ fold change >2) great differences were observed.

*5) There are concerns about the accuracy of the precursor frequency estimates. Caveats should be more fully discussed. Also discuss more fully the significance of the estimates.*

We agree with the reviewers that the method used to estimate MR1T cell precursor frequency has intrinsic limitations, being based on the assumption that the differences in the percentage of CD69^high^CD137^+^ activated T cells after stimulation with APCs expressing or lacking MR1 approximate the frequency of MR1T cells. We were aware that some of the T cells included in this gate might not be true MR1T cells. For this reason, we performed a second type of frequency evaluation in which single cell cloning was made after sorting of CD69^high^CD137^+^ T cells stimulated overnight with MR1-expressing A375 cells. The analysis of the clones showed that the sorted CD69^high^CD137^+^ T cells were enriched in MR1T cells. This second approach revealed similar frequency estimates.

Other methods are needed to exactly define MR1T cell frequencies. The use of MR1- tetramers is hampered by current lack of knowledge of nature and repertoire size of MR1T stimulatory antigens.

In conclusion, we are confident that the frequency estimates we obtained, although being approximations, indicated that MR1T cells are readily detectable ex vivoin all the donors analyzed without the need of in vitroexpansion, and therefore suggested that they are not a rare T cell population in the blood of healthy individuals. The revised manuscript has been modified (Results, subsection “MRI-reactive T cells are readily detectable in healthy individuals”, last paragraph and Discussion, second paragraph) to thoroughly discuss these issues.

*6) We expect that the TCR sequences will be submitted if the manuscript is published.*

The MR1T TCR sequences are subject of a pending patent application. We will make them available for collaborative studies.

[Editors' note: further revisions were requested prior to acceptance, as described below.]

*The manuscript has been improved but there is one quite significant issue that must be addressed before acceptance, as outlined below:*

*Although the authors have responded to most of the reviewers' comments, the reviewers and Editors are particularly concerned that the sequences of the TCR are being withheld because of a patent application. The authors' identification of the TCRs is based on the sequences, and those data are the primary data on which this conclusion is drawn. Also, the statement that they will make sequences available "for collaboration" is unacceptable. The sequences must be available on publication and deposited in a public database. These conditions are non-negotiable.*

As requested we have added the sequences information. In the subsection “TCR gene analysis of MR1T cell clones”, we provide the GenBank accession numbers where all the TCR sequences reported in Table 1 will be available.

**References**

1) Richards DM, Kyewski B, Feuerer M. 2016. Re-examining the Nature and Function of Self-Reactive T cells. Trends Immunol 37: 114-25.

2) de Jong A, Cheng TY, Huang S, Gras S, Birkinshaw RW, Kasmar AG, Van Rhijn I, Pena-Cruz V, Ruan DT, Altman JD, Rossjohn J, Moody DB. 2014. CD1a- autoreactive T cells recognize natural skin oils that function as headless antigens. Nat Immunol 15: 177-85.

3) de Jong A, Pena-Cruz V, Cheng TY, Clark RA, Van Rhijn I, Moody DB. 2010. CD1a-autoreactive T cells are a normal component of the human alphabeta T cell repertoire. Nat Immunol 11: 1102-9.

4) de Lalla C, Lepore M, Piccolo FM, Rinaldi A, Scelfo A, Garavaglia C, Mori L, De Libero G, Dellabona P, Casorati G. 2011. High-frequency and adaptive-like dynamics of human CD1 self-reactive T cells. Eur J Immunol 41: 602-10.

5) Kronenberg M, Rudensky A. 2005. Regulation of immunity by self-reactive T cells.

Nature 435: 598-604.

6) Le Bourhis L, Martin E, Peguillet I, Guihot A, Froux N, Core M, Levy E, Dusseaux M, Meyssonnier V, Premel V, Ngo C, Riteau B, Duban L, Robert D, Huang S, Rottman M, Soudais C, Lantz O. 2010. Antimicrobial activity of mucosal-associated invariant T cells. Nat Immunol 11: 701-8.

7) Kjer-Nielsen L, Patel O, Corbett AJ, Le Nours J, Meehan B, Liu L, Bhati M, Chen Z, Kostenko L, Reantragoon R, Williamson NA, Purcell AW, Dudek NL, McConville MJ, O'Hair RA, Khairallah GN, Godfrey DI, Fairlie DP, Rossjohn J, McCluskey J. 2012. MR1 presents microbial vitamin B metabolites to MAIT cells. Nature 491: 717-23.

8) Corbett AJ, Eckle SB, Birkinshaw RW, Liu L, Patel O, Mahony J, Chen Z, Reantragoon R, Meehan B, Cao H, Williamson NA, Strugnell RA, Van Sinderen D, Mak JY, Fairlie DP, Kjer-Nielsen L, Rossjohn J, McCluskey J. 2014. T-cell activation by transitory neo-antigens derived from distinct microbial pathways. Nature 509: 361-5.

9) Bacher P, Scheffold A. 2013. Flow-cytometric analysis of rare antigen-specific T cells. Cytometry A 83: 692-701.

10) Montoya CJ, Pollard D, Martinson J, Kumari K, Wasserfall C, Mulder CB, Rugeles MT, Atkinson MA, Landay AL, Wilson SB. 2007. Characterization of human invariant natural killer T subsets in health and disease using a novel invariant natural killer T cell-clonotypic monoclonal antibody, 6B11. Immunology 122: 1-14.

11) Tang XZ, Jo J, Tan AT, Sandalova E, Chia A, Tan KC, Lee KH, Gehring AJ, De Libero G, Bertoletti A. 2013. IL-7 licenses activation of human liver intrasinusoidal mucosal-associated invariant T cells. J Immunol 190: 3142-52

12) Mardiney M, 3rd, Brown MR, Fleisher TA. 1996. Measurement of T-cell CD69 expression: a rapid and efficient means to assess mitogen- or antigen-induced proliferative capacity in normals. Cytometry 26: 305-10.

13) Sun S, Zhang X, Tough DF, Sprent J. 1998. Type I interferon-mediated stimulation of T cells by CpG DNA. J Exp Med 188: 2335-42.

14) Tough DF, Sun S, Sprent J. 1997. T cell stimulation in vivo by lipopolysaccharide (LPS). J Exp Med 185: 2089-94.

15) Wolfl M, Kuball J, Ho WY, Nguyen H, Manley TJ, Bleakley M, Greenberg PD. 2007. Activation-induced expression of CD137 permits detection, isolation, and expansion of the full repertoire of CD8+ T cells responding to antigen without requiring knowledge of epitope specificities. Blood 110: 201-10.

16) Wolfl M, Kuball J, Eyrich M, Schlegel PG, Greenberg PD. 2008. Use of CD137 to study the full repertoire of CD8+ T cells without the need to know epitope specificities. Cytometry A 73: 1043-9.

17) Choi BK, Lee SC, Lee MJ, Kim YH, Kim YW, Ryu KW, Lee JH, Shin SM, Lee SH, Suzuki S, Oh HS, Kim CH, Lee DG, Hwang SH, Yu EM, Lee IO, Kwon BS. 2014. 4-1BB-based isolation and expansion of CD8+ T cells specific for self-tumor and non-self-tumor antigens for adoptive T-cell therapy. J Immunother 37: 225-36.

18) Huang S, Gilfillan S, Cella M, Miley MJ, Lantz O, Lybarger L, Fremont DH, Hansen TH. 2005. Evidence for MR1 antigen presentation to mucosal-associated invariant T cells. J Biol Chem 280: 21183-93.

19) Chua WJ, Kim S, Myers N, Huang S, Yu L, Fremont DH, Diamond MS, Hansen TH. 2011. Endogenous MHC-related protein 1 is transiently expressed on the plasma membrane in a conformation that activates mucosal-associated invariant T cells. J Immunol 186: 4744-50.

20) Becattini S, Latorre D, Mele F, Foglierini M, De Gregorio C, Cassotta A, Fernandez B, Kelderman S, Schumacher TN, Corti D, Lanzavecchia A, Sallusto F. 2015. T cell immunity. Functional heterogeneity of human memory CD4(+) T cell clones primed by pathogens or vaccines. Science 347: 400-6.